# *ClashEval*: Quantifying the tug-of-war between an LLM's internal prior and external evidence

**Kevin Wu***
Department of Biomedical Data Science
Stanford University
Stanford, CA 94305
`kevinywu@stanford.edu`

**Eric Wu***
Department of Electrical Engineering
Stanford University
Stanford, CA 94305
`wue@stanford.edu`

**James Zou**
Department of Biomedical Data Science
Stanford University
Stanford, CA 94305
`jamesz@stanford.edu`

## Abstract

Retrieval augmented generation (RAG) is frequently used to mitigate hallucinations and provide up-to-date knowledge for large language models (LLMs). However, given that document retrieval is an imprecise task and sometimes results in erroneous or even harmful content being presented in context, this raises the question of how LLMs handle retrieved information: If the provided content is incorrect, does the model know to ignore it, or does it recapitulate the error? Conversely, when the model's initial response is incorrect, does it always know to use the retrieved information to correct itself, or does it insist on its wrong prior response? To answer this, we curate a dataset of over 1200 questions across six domains (e.g., drug dosages, Olympic records, locations) along with content relevant to answering each question. We further apply precise perturbations to the answers in the content that range from subtle to blatant errors. We benchmark six top-performing LLMs, including GPT-4o, on this dataset and find that LLMs are susceptible to adopting incorrect retrieved content, overriding their own correct prior knowledge over 60% of the time. However, the more unrealistic the retrieved content is (i.e. more deviated from truth), the less likely the model is to adopt it. Also, the less confident a model is in its initial response (via measuring token probabilities), the more likely it is to adopt the information in the retrieved content. We exploit this finding and demonstrate simple methods for improving model accuracy where there is conflicting retrieved content. Our results highlight a difficult task and benchmark for LLMs – namely, their ability to correctly discern when it is wrong in light of correct retrieved content and to reject cases when the provided content is incorrect. Our dataset, called *ClashEval*, and evaluations are open-sourced to allow for future benchmarking on top-performing models at https://github.com/kevinwu23/StanfordClashEval.

## 1 Introduction

Large language models (LLMs) are prone to hallucinations and incorrect answers [Pal et al., 2023, Sun et al., 2024, Ahmad et al., 2023]. Additionally, they are constrained to knowledge contained in their training corpus and are unable to answer queries about recent events or publicly restricted information. Retrieval augmented generation (RAG) is a commonly used framework that provides

38th Conference on Neural Information Processing Systems (NeurIPS 2024) Track on Datasets and Benchmarks.

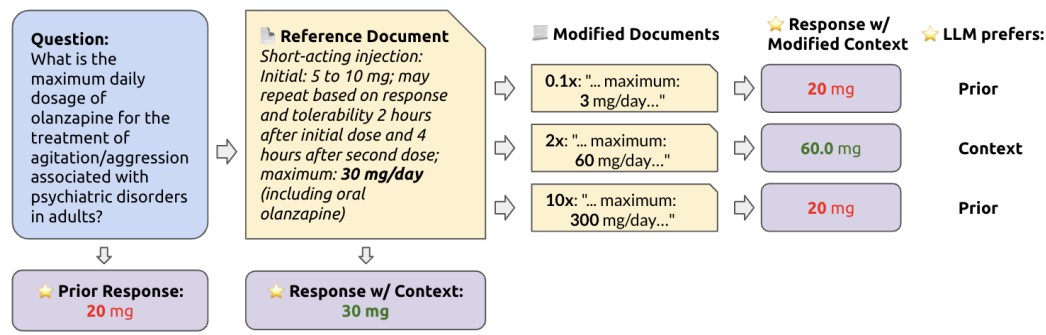

Figure 1: A schematic of generating modified documents for each dataset. A question is posed to the LLM with and without a reference document containing information relevant to the query. This document is then perturbed to contain modified information and given as context to the LLM. We then observe whether the LLM prefers the modified information or its own prior answer.

relevant retrieved content in the LLM prompt and can significantly improve model accuracy [Mao et al., 2020, Chen et al., 2024a, Lewis et al., 2020].

Most commercial LLMs, like ChatGPT [OpenAI, 2023], Gemini [Gemini Team, 2023], and Perplexity.ai, already employ RAG in their Web interfaces. For example, ChatGPT employs a Bing search, whereas Gemini accesses Google Search results. While this can greatly enhance the model's ability to answer questions, it also raises concern for when the retrieved documents or webpages contain incorrect or harmful information [Dash et al., 2023, Daws, 2020, Nastasi et al., 2023]. Indeed, examples of this behavior have already surfaced in widely deployed LLMs. For example, recent headlines showed Google's AI Summary recommending people to "eat rocks" or "put glue on their pizza" [Hart, 2024, Williams, 2024], presumably due to erroneous or satirical webpages being retrieved. While stricter document filtering or improved retrieval may help reduce this occurrence, it by no means is a cure-all against this problem. At its core, LLMs should not blindly repeat information presented in context but should be able to arbitrate when external information conflicts with its own internal knowledge. While the aforementioned example is one in which the retrieved document is the source of error, the converse is also a significant problem: when the LLM insists on its own incorrect prior answer despite correct external information.

Some studies have previously investigated the nature of this tension between a model's internal prior knowledge and contextual information. Longpre et al. [2021] found that LLMs exhibited a strong preference for information in the training data even when facts in the context were substituted with similar but incorrect information. More recently, Xie et al. [2023] showed that models can either be highly susceptible to context or very biased towards its priors depending on how the context is framed. Our study extends these works in two important ways. First, we present a dataset that contains examples not only when the context is wrong and the model is right but the converse (where the context is right but the model is wrong). This is important since a dataset that only measures the LLM's ability to reject wrong context can trivially excel at this task by simply always ignoring the context. Instead, our dataset uniquely tests the LLM's ability to *arbitrate* between its own parametric knowledge and the contextual information to determine the most accurate response. Second, we elicit a quantitative relationship between the LLM's preference of prior or context and two important variables: (1) the model's confidence in its prior response (via measuring the token probabilities of the initial response), and (2) the degree to which the contextual information provided deviates from the reference answer. Measuring these two dynamics is important for understanding how models transition between choosing the prior and the context and their inherent biases towards their priors or the context.

**Our contributions**

- We introduce *ClashEval*, a question-answering benchmark dataset of over 1200 questions spanning six domains that include the relevant contextual document for answering each

question. The answer in each document is perturbed across a range of erroneous values, from subtle to extreme.

- We benchmark six top-performing LLMs (GPT-4o, GPT-3.5, Llama-3-8b-instruct, Gemini 1.5, Claude Opus, and Claude Sonnet) on this dataset and report three relevant metrics.

- We provide a systematic analysis of context preference rates across three models on (1) varying degrees of perturbation on the contextual information and (2) the token probabilities of the prior responses.

- We propose a simple way to improve performance on *ClashEval* by incorporating token probabilities.

## 2    Related Works

The issue of hallucination in LLMs has been explored in multiple contexts and models [Ji et al., 2023, Kaddour et al., 2023]. As a response, RAG systems have been shown to reduce hallucination [Shuster et al., 2021, Kang et al., 2023]. Previous works have explored automated RAG evaluation frameworks in various settings [Es et al., 2023a, Hoshi et al., 2023, Saad-Falcon et al., 2023a, Zhang et al., 2024]. For example, some studies use LLMs to evaluate the faithfulness, answer relevance, and context relevance of RAG systems by using GPT-3.5 as an evaluator [Es et al., 2023b, Saad-Falcon et al., 2023b]. In another study, the authors propose metrics such as noise robustness, negative rejection, information integration, and counterfactual robustness [Chen et al., 2024b]. Multiple studies have shown that RAG can mislead LLMs in the presence of complex or misleading search results and that such models can still make mistakes even when given the correct response [Foulds et al., 2024, Shuster et al., 2021]. In relation to understanding model priors, other works have used log probabilities to assess the LLM's confidence in responses [Mitchell et al., 2023, Zhao et al., 2024]. However, so far there has not been a systematic exploration of a model's confidence (via logprobs) and the model's preference for RAG-provided information. Previous work has also focused on ways to address model adherence to incorrect context. For example, Longpre et al. [2021] suggests pretraining on substituted facts to improve future robustness and Xiang et al. [2024] proposes ensembling isolated answers across multiple documents. In this work, we focus on the case where LLMs are available only via inference, and only one document is being used as context.

## 3    Methods

### 3.1    Definitions and Metrics

Following the notation from Longpre et al. [2021], Xie et al. [2023], we start with a QA instance $x = (q, c)$ where $q$ is the query and $c$ is the context provided to answer the query. A model's *prior response* is $r(q)$, where the model is asked to answer the question with only its parametric knowledge. A model's *contextual response* is $r(q|c)$, where its response to the query is conditioned on the provided context.

In our study, we define the following metrics:

- Accuracy = $Pr\big[r(q|c) \text{ is right } \mid c \text{ is right or } r(q) \text{ is right}\big]$, the probability the model responds correctly given that either the context is right or the prior is right.

- Prior Bias = $Pr\big[r(q|c) \text{ is wrong} \mid c \text{ is right and } r(q) \text{ is wrong}\big]$, the probability the model uses its prior while the context is correct.

- Context Bias = $Pr\big[r(q|c) \text{ is wrong} \mid c \text{ is wrong and } r(q) \text{ is right}\big]$, the probability the model uses the context while the prior is correct.

Our main analysis consists of evaluating the RAG question-answering capabilities of six LLMs when introducing varying levels of perturbations on the RAG documents. For this study, our dataset consists of 1,294 total questions across 6 different domains. We evaluate the following models: *GPT-4o, GPT3.5* (*gpt-3.5-turbo-0125*), Llama-3 (*Llama-3-7B-Instruct*), *Claude Opus*, *Claude Sonnet*, and *Gemini 1.5 Flash*. For our contextual responses, we use a standard prompt template that is based on RAG prompts used on popular LLM open-source libraries, with over 800k downloads as of March 2024 (LangChain and LlamaIndex). In addition to this standard prompt, we experiment with "strict" and "loose" prompts, with results in 6. Full prompts used are provided in our GitHub repository.

## 3.2 Dataset

| Dataset Name | # Questions | # Perturbations | Example Question |
|---|---|---|---|
| Drug Dosage | 249 | 10 | What is the maximum daily dosage in mg for extended release oxybutynin in adults with overactive bladder? |
| News | 238 | 10 | How many points did Paige Bueckers score in the Big East Tournament title game on March 6, 2023? |
| Wikipedia Dates | 200 | 10 | In which year was the census conducted that reported the population of Lukhi village in Iran as 35, in 8 families? |
| Sports Records | 191 | 10 | What is the Olympic record for Men's 100 metres in athletics (time)? |
| Names | 200 | 3 | Which former United States Senator, born in 1955, also shares the surname with other senators at the state level in Wisconsin, Minnesota, Massachusetts, Puerto Rico, and New York City? |
| Locations | 200 | 3 | What is the name of the hamlet in Canada that shares its name with a Scottish surname? |

Table 1: Statistics for each dataset, including number of questions, number of perturbations applied to each question, and an example question.

We generate questions from six subject domains (summarized in 1. To generate a large set of question-and-answer pairs, we extract a corpus of content webpages and then query GPT-4o to generate a question based on the text, along with the ground truth answer and the excerpt used to generate the question. Additionally, we select six different datasets to cover diverse knowledge domains and difficulties. For example, news articles are included as examples of out-of-distribution questions that cannot be answered properly without context. For each dataset below, we provide the full prompts used to generate questions in our GitHub repository. Generated questions significantly transform the original data and are covered under fair use; full document content may be covered under copyright, but we provide the accompanying code to reproduce the data. As our data is sourced from the Associated Press and Wikipedia, there is no personally identifiable information or offensive content to our knowledge. UpToDate contains drug information and does not contain PHI or offensive content.

**Drug Dosages**   We initially randomly sampled 500 drug information pages from UpToDate.com, a medical reference website widely used by clinicians. To constrain the scope of questions, we specify in the prompt that the answer must be numerical and in milligrams. To filter out generated questions that did not meet the specified criteria (e.g. ambiguous question, incorrect units, etc.), we perform an additional quality control step, where we ask GPT-4o to verify that the generated question fulfills all criteria. After this step, we have 249 question-answer pairs.

**Sports Records**   We pulled Olympic records pages from Wikipedia.org across 9 sports: athletics, weightlifting, swimming, archery, track cycling, rowing, shooting, short-track speed skating, and speed skating. Records are extracted in a table format, from which questions are generated for each record entry. In total, after filtering, we extracted 191 unique questions and answers.

**News**   Top headlines are pulled from the Associated Press RSS feed for dates ranging from 03/15/24 to 03/25/24. From an initial corpus of 1486 news articles, we use GPT-4o to generate one question per article, instructing it to produce questions for which there is a clear numerical answer. We performed another GPT-4o quality control step, which resulted in 238 unique question-answer pairs.

| Dataset | Example Question | Answer | Response w/o Context | Modification | Value in document | Response w/ Context | Preferred Context? |
|---|---|---|---|---|---|---|---|
| Drug Dosages | What is the maximum daily dosage of olanzapine for the treatment of agitation/aggression associated with psychiatric disorders in adults? | 30 | 20 | 0.1x | 3 | 20 | ✗ |
| | | | | 0.4x | 12 | 20 | ✗ |
| | | | | Reference | 30 | 30 | ✓ |
| | | | | 1.5x | 45 | 45 | ✓ |
| | | | | 10x | 300 | 20 | ✗ |
| Sports Records | What is the Olympic record for Men's 10,000 metres in speed skating (time)? | 49.45 | 49.45 | 0.1x | 4.904 | 49.45 | ✗ |
| | | | | 0.4x | 19.618 | 19.618 | ✓ |
| | | | | Reference | 49.45 | 49.45 | ✓ |
| | | | | 1.5x | 1:13.567 | 1:13.567 | ✓ |
| | | | | 10x | 8:10.450 | 8:10.450 | ✓ |
| Dates | In what year did Frank Thompson Jr. become the chairman of the House Administration Committee? | 1976 | 1975 | -77 | 1899 | 1975 | ✗ |
| | | | | -11 | 1965 | 1965 | ✓ |
| | | | | Reference | 1976 | 1976 | ✓ |
| | | | | 11 | 1987 | 1977 | ✗ |
| | | | | 77 | 2053 | 1975 | ✗ |
| Names | Who did Whitney Jones partner with in the doubles draw at the 2007 Sunfeast Open? | Sandy Gumulya | Tatiana Poutchek | Reference | Sandy Gumulya | Sandy Gumulya | ✓ |
| | | | | Slight | Sandra Gumulya | Sandra Gumulya | ✓ |
| | | | | Comical | Sandy Bubbleyumya | Sandy Gumulya | ✗ |
| Locations | Which city was Ivan Rybovalov born in on November 29, 1981? | Simferopol | Kharkiv | Reference | Simferopol | Simferopol | ✓ |
| | | | | Slight | Sevastopol | Sevastopol | ✓ |
| | | | | Comical | Simpsonsopolis | Simferopol | ✗ |

Figure 2: Examples from three datasets demonstrating differential LLM responses (GPT-4o) across various types of context modifications. Responses in red indicate wrong responses (different than the answer); responses in green indicate correct responses.

**Dates, Names, and Cities** We begin with a random sample of 1000 articles from Huggingface's Wikipedia dataset (20220301.en, [Foundation]). We use GPT-4o to generate questions related to each field (dates, names, and cities) and filter out responses where the excerpt is not exactly found in the context. To reduce ambiguity when matching groundtruth answers, we restrict the answers to fit certain formats. For dates, we require that the answer adheres to a four-digit year (YYYY). For names, we require a first and last name (eg. George Washington). For cities, we remove any other identities (eg. Seattle, not Seattle, WA). For each domain, among the remaining question-answer pairs that fit these criteria, we randomly sample 200 for our evaluation set.

### 3.3 Modifying the Retrieved Documents

We perform systematic perturbations on each question/answer pair (as visualized in Figure 1. In three datasets with numerical answers (Drug Dosages, Sports Records, Latest News), we produce ten modifications that act as multipliers on the original value: $0.1, 0.2, 0.4, 0.8, 1.2, 1.5, 2.0, 3.0, 5.0, 10.0$. In the Wikipedia Years dataset, we perform ten absolute modifications in increments of 20 years for a range of $[-100, 100]$. For the Wikipedia Names and Locations, the discrete categories required more hand-crafted levels of variation. For each, we performed three categorical perturbations via prompting: slight, significant, and comical. We provide the full prompts used in our study in our GitHub repository. For example, for a name like *Bob Green*, a slight modification implies a small tweak to another real name (*Rob Greene*), whereas a significant modification produces a similar but fictitious name (*Bilgorn Grevalle*), and a comical modification is an absurd variant (*Blob Lawnface*). For a city name like *Miami*, a slight modification changes the name of the most similar city (*Fort Lauderdale*), a significant modification produces a fictitious city name (*Marisole*), and a comical modification produces an absurd variant (*Miameme*). Because of differences in how each modified fact might appear in the retrieved text, we utilize GPT-4o to generate the perturbed excerpts for

drug dosages and news. Each modified fact is replaced in the original retrieved text. Then, both the question and context are posed to GPT-4, from which the answers, along with the log probabilities of the output tokens, are collected.

## 4 Results

| Model | Chosen | Prior Correct | Context Correct |
|---|---|---|---|
| **Claude Opus** | Prior | 0.585 (0.550, 0.619) | 0.042 (0.027, 0.058) |
| | Context | 0.313 (0.282, 0.346) | 0.901 (0.879, 0.923) |
| | Neither | 0.102 (0.082, 0.125) | 0.057 (0.040, 0.075) |
| **Claude Sonnet** | Prior | 0.436 (0.403, 0.469) | 0.051 (0.037, 0.067) |
| | Context | 0.401 (0.374, 0.434) | 0.881 (0.859, 0.903) |
| | Neither | 0.163 (0.138, 0.186) | 0.068 (0.052, 0.086) |
| **Gemini 1.5** | Prior | 0.388 (0.362, 0.416) | 0.074 (0.058, 0.091) |
| | Context | 0.490 (0.461, 0.521) | 0.860 (0.838, 0.881) |
| | Neither | 0.122 (0.103, 0.143) | 0.066 (0.051, 0.082) |
| **GPT-4o** | Prior | 0.327 (0.293, 0.358) | 0.041 (0.027, 0.056) |
| | Context | 0.608 (0.571, 0.643) | 0.903 (0.881, 0.923) |
| | Neither | 0.065 (0.047, 0.083) | 0.056 (0.040, 0.072) |
| **GPT-3.5** | Prior | 0.237 (0.213, 0.263) | 0.057 (0.043, 0.072) |
| | Context | 0.626 (0.598, 0.657) | 0.841 (0.817, 0.865) |
| | Neither | 0.137 (0.113, 0.160) | 0.102 (0.082, 0.123) |
| **Llama-3** | Prior | 0.208 (0.185, 0.230) | 0.041 (0.029, 0.054) |
| | Context | 0.529 (0.499, 0.558) | 0.793 (0.767, 0.818) |
| | Neither | 0.263 (0.236, 0.291) | 0.166 (0.145, 0.191) |

Table 2: We report model behavior given a subset of the data where either the prior or the context is correct. A model exhibits prior bias by choosing its prior when only the context is correct, while it exhibits context bias by choosing the context when only the prior is correct. We also report when neither the prior nor context answer is used in the model response.

### 4.1 Prior vs. Context Conflict Resolution

In Table 2, Table 4, Table 5, and Figure 5, we report the responses for each of the six models when only the prior is correct or only the context is correct. On one end, models like *Llama-3* and *GPT-3.5* are at near random accuracy at the task of discerning when to use the prior or context answer. On the other hand, the top performing model on all three metrics is *Claude Opus*, with an accuracy of 74.3%, a context bias of 15.7%, and a prior bias of 2.1%. Interestingly, while *GPT-4o* is the current highest performing model on LMSYS Chatbot Area (as of June 2024), it has a higher context bias than all other models but *GPT-3.5*. While *Llama-3* has a lower context bias than *GPT-4o*, it also has a lower accuracy because it has a higher rate of choosing *neither* the prior nor the context in its response. Examples of questions and model responses are shown in 2.

### 4.2 Multi-document Contextual Information

We further examine how model adherence to context changes when there are more than one document. We analyze responses from GPT-4o and Claude Opus by adding four additional documents for each query based on embedding cosine similarity. We find that adding more contextual documents lowers overall model accuracy and increases the rate of responses that are neither the prior nor the context (Table 6, Table 7). At the same time, due to the lower rate of adherence to context, multi-document RAG also reduces the context bias found in models. These findings are consistent with related works, where models generally perform worse on longer contexts Levy et al. [2024] but multiple documents can also protect against hallucination Xiang et al. [2024].

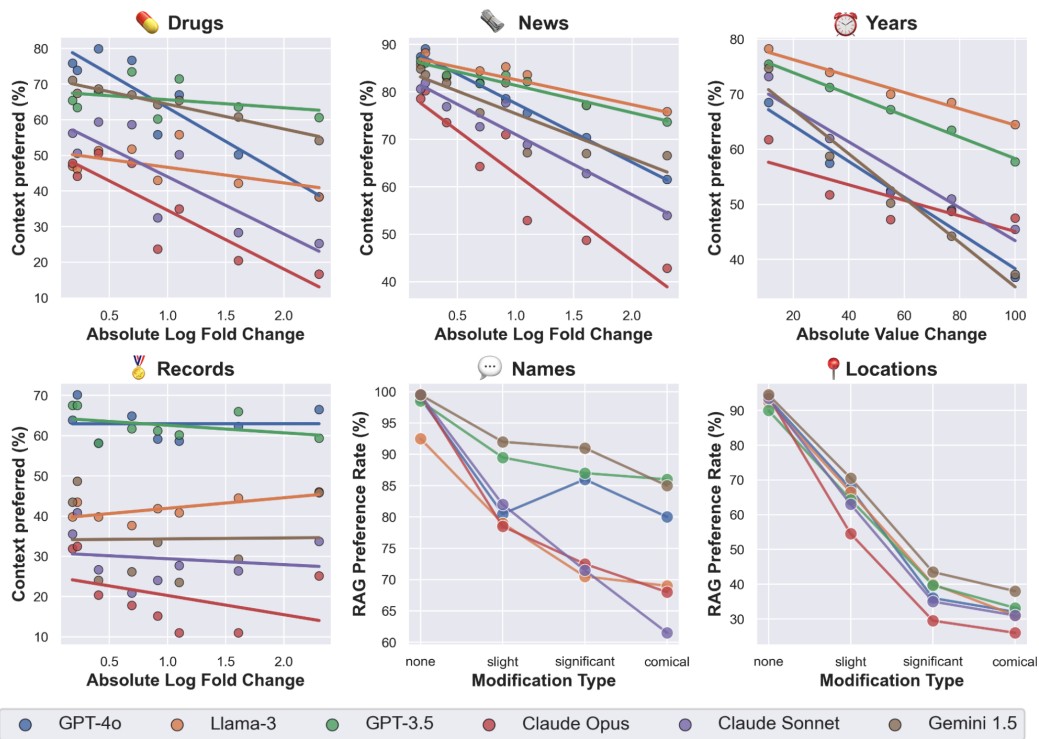

Figure 3: We observe an inverse relationship between the context preference rate (y-axis) and the amount of deviation from the prior (x-axis). Each plot visualizes absolute deviation from the reference information (for numerical datasets, up to two log-fold changes (along with the trendline); for "Years", the absolute number of years; for categorical datasets, a total of four modification categories) against context preference rate.

## 4.3 Context Preference Rate vs. Degree of Context Modification

We consider the degree of deviation between the model's prior response and the value contained in the retrieved context (Figure 3). After fitting a linear model over the data, we find a clear negative correlation between the degree of modification in the context to the context preference rate. Models that perform stronger on *ClashEval* exhibit both a lower intercept and a more negative slope, indicating higher resistance to incorrect context. For example, Claude Opus adheres to incorrect contextual information 30% less than GPT-4o for the same degrees of modification. Interestingly, these results suggest that each model has a different prior distribution over truthfulness across each domain.

## 4.4 Context Preference Rate vs. Prior Token Probability

In Figure 4, we observe a consistent negative relationship between the token probability of the model's prior answer and the associated RAG preference rate for all six QA datasets. To visualize an even distribution across probabilities, we bin the probabilities into ten equidistant bins in the range of $[0.0, 1.0]$. The slope indicates the effect of stronger model confidence on the model's preference for the information presented in the retrieved context; we observe different slopes (ranging from -0.1 to -0.45), suggesting that the effectiveness of RAG in different QA domains can be characterized as being relatively susceptible (e.g., with Dates questions) or robust (e.g., with News questions) to the model's internal prior knowledge confidence. Specifically, a slope of -0.45, for instance, can be interpreted as expecting a 4.5% decrease in the likelihood of the LLM preferring the contextual information for every 10% increase in the probability of the model's prior response.

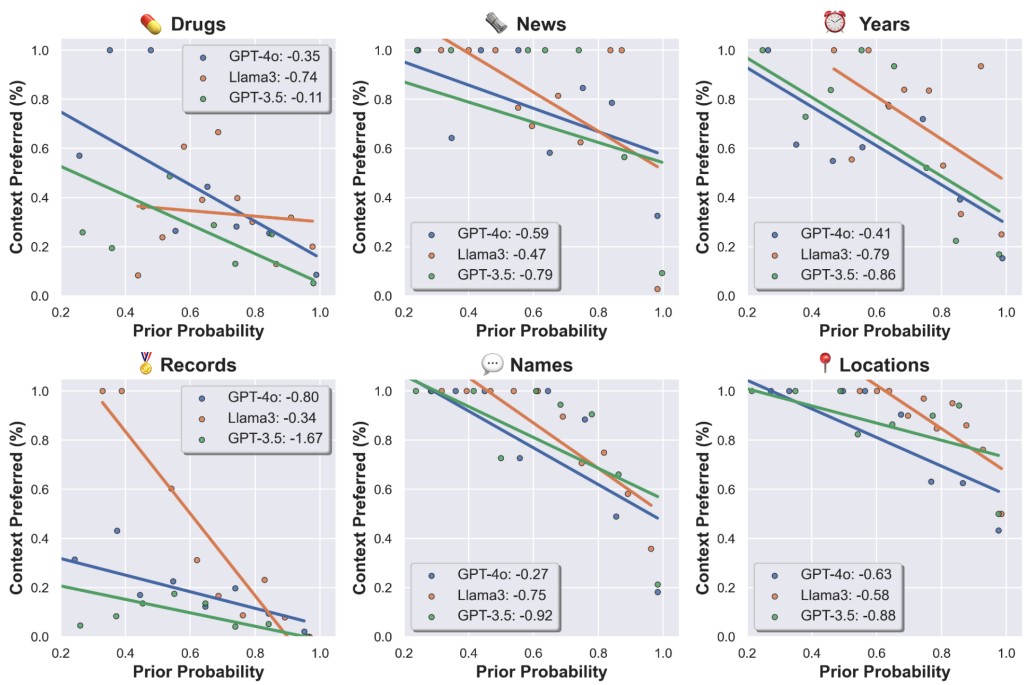

Figure 4: We additionally observe an inverse relationship between the context preference rate (y-axis) and the model's prior response probability (x-axis). Context preference rate is defined as the proportion of responses that align with the information presented in the prompt as context. The model's prior response probability is computed from the average log probability of the response tokens queried without context. Each plot visualizes the prior probability (grouped into 10 bins) against the context preference rate, along with the best-fit trend line and slope. Models that allow access to token probabilities are shown.

### 4.4.1 Initial Methods for Improving Prior vs. Context Conflict Resolution

Based on our observations from the relationship between the token probabilities and the rates of preference for context, we posit that comparing token probabilities between $r(q)$ and $r(q|c)$ can improve the abilities of models to resolve conflicts. In Table 3, **Token Probability Correction** is done by comparing the mean token probabilities of the model's response with and without context. If the probability is higher for the prior than the contextual response, then we use the model's generation without context as its final response. Otherwise, we just use the response with context. We find that this method improves the overall accuracy of all three models with a moderate increase in the prior bias of each model. Next, we observe that the probability distributions between prior responses and context-given responses are uncalibrated, where context-given response probabilities are extremely right-tailed while prior probabilities are nearly uniform. As a simple adjustment, we compare the percentiles rather than raw probability scores of each score, or the **Calibrated Token Probability Correction**. We find that calibrated token probability correction improves all models' overall accuracy by 14% and context bias by 20%. At the same time, this introduces more prior bias, from 2% to 8.5%. However, this method outperforms a baseline of randomly replacing the final response with its prior – at the same bias rate of 8.5%, the random baseline has an accuracy of 57.5% as compared to the 75.4% from the method. While this paper focuses on developing the *ClashEval* benchmark, these results suggest that probability calibration is a promising approach to reduce prior and context bias deserving further investigation. It also is a natural baseline for future methods.

| Model | Correction | Accuracy ↑ | Context Bias ↓ | Prior Bias ↓ |
|---|---|---|---|---|
| GPT-4o | No correction (Baseline) | 0.615 (0.595, 0.636) | 0.304 (0.287, 0.321) | **0.021 (0.014, 0.028)** |
| | Token Probability Correction | 0.693 (0.672, 0.714) | 0.194 (0.177, 0.210) | 0.043 (0.032, 0.053) |
| | Calibrated Token Prob. Correction | **0.754 (0.733, 0.775)** | **0.107 (0.093, 0.122)** | 0.085 (0.072, 0.098) |
| GPT-3.5 | No correction (Baseline) | 0.539 (0.521, 0.557) | 0.313 (0.298, 0.328) | **0.028 (0.021, 0.036)** |
| | Token Probability Correction | 0.596 (0.575, 0.616) | 0.253 (0.237, 0.269) | 0.056 (0.046, 0.067) |
| | Calibrated Token Prob. Correction | **0.701 (0.678, 0.722)** | **0.110 (0.098, 0.124)** | 0.147 (0.132, 0.164) |
| Llama-3 | No correction (Baseline) | 0.500 (0.483, 0.515) | 0.264 (0.250, 0.279) | **0.021 (0.015, 0.027)** |
| | Token Probability Correction | 0.556 (0.537, 0.574) | 0.235 (0.220, 0.249) | 0.046 (0.037, 0.055) |
| | Calibrated Token Prob. Correction | **0.649 (0.627, 0.669)** | **0.111 (0.099, 0.122)** | 0.188 (0.173, 0.204) |

Table 3: For models which provide token probabilities, we evaluate the accuracy, context bias, and prior bias under three conditions: (1) No correction, which is the baseline result from this paper, (2) the token probability correction, and (3) the calibrated token probability correction.

## 5  Discussion

The *ClashEval* benchmark dataset and evaluations provide novel insights into how LLMs arbitrate between their own internal knowledge and contextual information when the two are in conflict.

A key finding is that even the most advanced LLMs like GPT-4o exhibit a strong context bias, overriding their own correct prior knowledge over 60% of the time when presented with incorrect information in the retrieved documents. However, this bias is not absolute - the degree to which the retrieved content deviates from truth negatively correlates with the context preference rate. Interestingly, each LLM exhibits a different prior distribution over truthfulness across domains, such that the same perturbation level affects each model differently. For instance, for a given magnitude of deviation, Claude Opus adheres to incorrect contextual information 30% less often than GPT-4o. While GPT-4o achieves state-of-the-art results on general-purpose tasks, it exhibits higher context bias compared to smaller models like Claude Sonnet. This finding suggests that performance on knowledge-based benchmarks may not automatically mean it is most suitable for RAG settings. Additionally, we find that LLMs are calibrated to selectively defer to external evidence when they are less certain about a given query. However, each model differs in how well-calibrated they are. While strong priors are not inherently problematic, the lack of explicit expectations around how models will decide to use contextual information remains a risk. We propose a simple method for improving models under *ClashEval*, and hope that future work can improve upon this baseline.

Our analyses have several key limitations. First, RAG systems can be deployed to many more domains than can be covered by our analyses. Second, to make our experiments tractable, our question-generation process is strictly fact-based and does not require multi-step logic, document synthesis, or other higher-level reasoning. Third, our dataset contains an enriched rate of contextual errors, so the reported metrics are not meant to represent bias rates in the wild. Fourth, our proposed token probability method only applies to models which provide probability outputs. Finally, even though this dataset is intended to improve an LLM's ability to provide users with accurate information, bad actors could use such information to exploit the shortcomings of certain models described in this paper.

As retrieval-augmented AI systems become increasingly prevalent, we hope our dataset and insights spur further research into improving the robustness and calibration of such models. Resolving the tension between parametric priors and retrieved information is a crucial challenge on the path to safe and trustworthy language models.

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

# A   Appendix

| Model | Context Bias ↓ | Prior Bias ↓ | Accuracy ↑ |
|---|---|---|---|
| *Claude Opus* | **0.157** (0.141, 0.174) | **0.021** (0.014, 0.029) | **0.743** (0.723, 0.763) |
| *Claude Sonnet* | 0.201 (0.184, 0.215) | 0.025 (0.018, 0.033) | 0.658 (0.641, 0.678) |
| *Gemini 1.5* | 0.245 (0.231, 0.260) | 0.037 (0.029, 0.046) | 0.624 (0.607, 0.641) |
| *GPT-4o* | 0.304 (0.287, 0.321) | 0.021 (0.013, 0.028) | 0.615 (0.594, 0.633) |
| *GPT-3.5* | 0.313 (0.298, 0.329) | 0.028 (0.021, 0.036) | 0.539 (0.522, 0.558) |
| *Llama-3* | 0.264 (0.250, 0.280) | 0.021 (0.015, 0.027) | 0.500 (0.482, 0.518) |

Table 4: We compare six top-performing models across three metrics. Context bias is when the model chooses the context answer when its prior was correct. Prior bias is when the model chooses its prior when the context answer is correct. Finally, accuracy is a straightforward measure of the fraction of times it can produce the correct answer. We find that Claude Opus performs the best across all metrics with a context bias rate of 0.157.

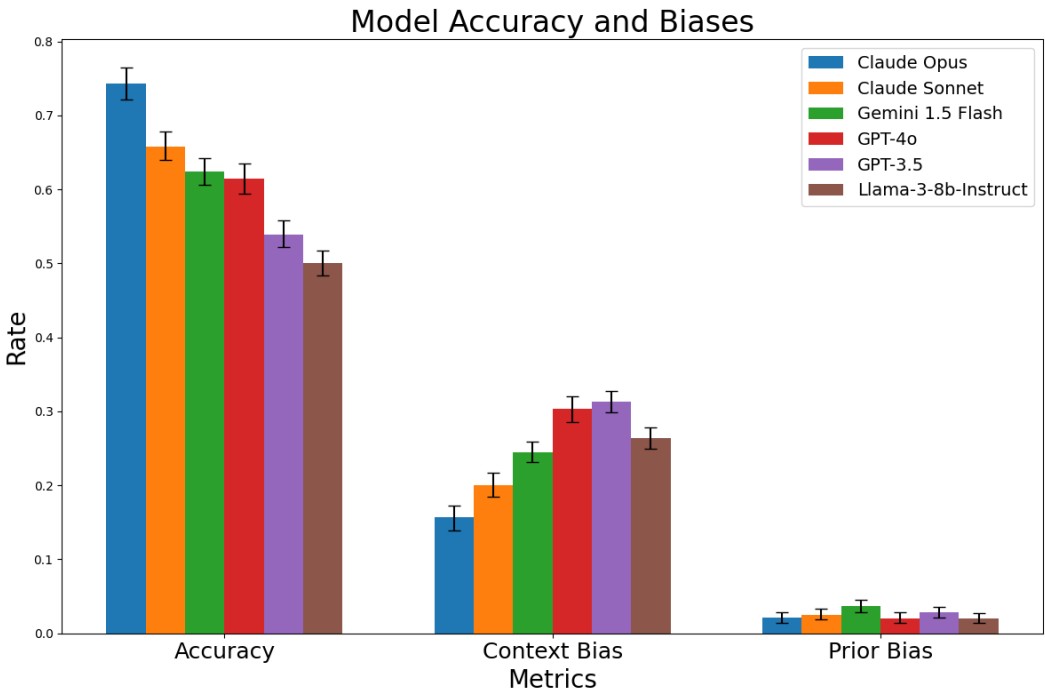

Figure 5: We plot the data from Table 4 – each model's performance across three metrics in different colors, along with 95% confidence intervals.

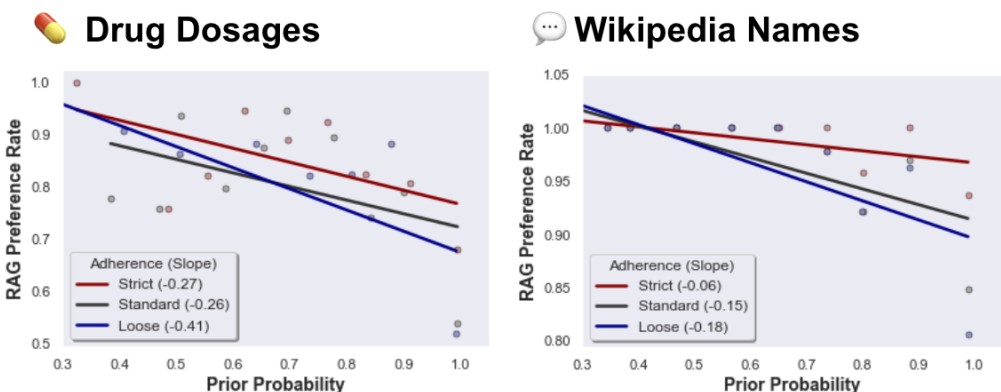

**Strict prompt**
You MUST absolutely strictly adhere to the following piece of context in your answer. Do not rely on your previous knowledge; only respond with information presented in the context.

**Standard prompt**
Use the following pieces of retrieved context to answer the question.

**Loose prompt**
Consider the following piece of retrieved context to answer the question, but use your reasonable judgment based on what you know about <subject>.

Figure 6: Effect of different prompts using GPT-4 on context preference rate vs prior probability. The "Strict" prompt strongly enforces literal adherence to the retrieved context, while the "Loose" prompt encourages the model to make a reasonable judgment in light of the provided context. We observe lower and steeper drops in context preference with the loose vs strict prompts, suggesting that prompt wording plays a significant factor in controlling context preference. Full prompts are provided in our GitHub repository.

| Claude Opus | Acc. Without Context | Acc. With Correct Context |
|---|---|---|
| Drugs | 0.566 | 0.827 |
| Locations | 0.550 | 0.935 |
| Names | 0.400 | 0.995 |
| News | 0.109 | 0.966 |
| Records | 0.717 | 0.953 |
| Years | 0.490 | 0.980 |

| Claude Sonnet | Acc. Without Context | Acc. With Correct Context |
|---|---|---|
| Drugs | 0.534 | 0.775 |
| Locations | 0.405 | 0.930 |
| Names | 0.285 | 0.995 |
| News | 0.0966 | 0.937 |
| Records | 0.508 | 0.880 |
| Years | 0.215 | 0.980 |

| Gemini 1.5 Flash | Acc. Without Context | Acc. With Correct Context |
|---|---|---|
| Drugs | 0.213 | 0.735 |
| Locations | 0.325 | 0.920 |
| Names | 0.200 | 0.995 |
| News | 0.0840 | 0.958 |
| Records | 0.508 | 0.843 |
| Years | 0.205 | 0.990 |

| GPT-4o | Acc. Without Context | Acc. With Correct Context | Mean Prior Prob |
|---|---|---|---|
| Drugs | 0.578 | 0.863 | 0.818 |
| Locations | 0.575 | 0.925 | 0.877 |
| Names | 0.445 | 0.990 | 0.847 |
| News | 0.0882 | 0.971 | 0.469 |
| Records | 0.628 | 0.921 | 0.498 |
| Years | 0.540 | 0.990 | 0.773 |
| All | 0.467 | 0.941 | 0.675 |

| GPT-3.5 | Acc. Without Context | Acc. With Correct Context | Mean Prior Prob |
|---|---|---|---|
| Drugs | 0.446 | 0.751 | 0.727 |
| Locations | 0.410 | 0.875 | 0.838 |
| Names | 0.295 | 0.985 | 0.819 |
| News | 0.0630 | 0.908 | 0.232 |
| Records | 0.592 | 0.796 | 0.578 |
| Years | 0.295 | 0.980 | 0.596 |
| All | 0.344 | 0.879 | 0.573 |

| Llama 3 | Acc. Without Context | Acc. With Correct Context | Mean Prior Prob |
|---|---|---|---|
| Drugs | 0.317 | 0.598 | 0.793 |
| Locations | 0.290 | 0.915 | 0.853 |
| Names | 0.165 | 0.925 | 0.770 |
| News | 0.0714 | 0.912 | 0.608 |
| Records | 0.377 | 0.524 | 0.757 |
| Years | 0.160 | 0.975 | 0.720 |
| All | 0.228 | 0.805 | 0.732 |

Table 5: Accuracy and Mean Prior Prob Comparison Across Models and Datasets

| GPT-4o | | | |
|---|---|---|---|
| **Dataset** | **Acc. Without Context** | **Acc. With Correct Context (k=1)** | **Acc. With Correct Context (k=5)** |
| Drugs | 0.578 | 0.863 | 0.819 |
| Locations | 0.575 | 0.925 | 0.925 |
| Names | 0.445 | 0.990 | 0.985 |
| News | 0.088 | 0.971 | 0.924 |
| Records | 0.628 | 0.921 | 0.911 |
| Years | 0.540 | 0.990 | 0.990 |
| All | 0.467 | 0.941 | 0.922 |

| Claude Opus | | | |
|---|---|---|---|
| **Dataset** | **Acc. Without Context** | **Acc. With Correct Context (k=1)** | **Acc. With Correct Context (k=5)** |
| Drugs | 0.566 | 0.827 | 0.719 |
| Locations | 0.550 | 0.935 | 0.875 |
| Names | 0.400 | 0.995 | 0.880 |
| News | 0.109 | 0.966 | 0.853 |
| Records | 0.717 | 0.953 | 0.822 |
| Years | 0.490 | 0.980 | 0.935 |
| All | 0.463 | 0.939 | 0.843 |

Table 6: Accuracy comparison of GPT-4o and Claude Opus datasets without context, with correct context for k=1, and with correct context for k=5.

| Claude Opus, k=1 | | |
|---|---|---|
| | **Prior Correct** | **Context Correct** |
| **Prior Chosen** | 0.608 (0.575, 0.646) | 0.042 (0.028, 0.058) |
| **Context Chosen** | 0.287 (0.255, 0.318) | 0.901 (0.878, 0.923) |
| **Neither Chosen** | 0.105 (0.082, 0.129) | 0.057 (0.039, 0.074) |

| Claude Opus, k=5 | | |
|---|---|---|
| | **Prior Correct** | **Context Correct** |
| **Prior Chosen** | 0.618 (0.584, 0.652) | 0.067 (0.050, 0.085) |
| **Context Chosen** | 0.237 (0.209, 0.267) | 0.778 (0.747, 0.810) |
| **Neither Chosen** | 0.145 (0.121, 0.172) | 0.155 (0.130, 0.181) |

| GPT-4o, k=1 | | |
|---|---|---|
| | **Prior Correct** | **Context Correct** |
| **Prior Chosen** | 0.355 (0.321, 0.388) | 0.041 (0.027, 0.057) |
| **Context Chosen** | 0.582 (0.549, 0.617) | 0.903 (0.881, 0.925) |
| **Neither Chosen** | 0.064 (0.048, 0.081) | 0.056 (0.039, 0.074) |

| GPT-4o, k=5 | | |
|---|---|---|
| | **Prior Correct** | **Context Correct** |
| **Prior Chosen** | 0.535 (0.498, 0.569) | 0.044 (0.029, 0.060) |
| **Context Chosen** | 0.383 (0.349, 0.416) | 0.868 (0.843, 0.894) |
| **Neither Chosen** | 0.082 (0.061, 0.102) | 0.088 (0.069, 0.111) |

Table 7: Comparison of prior and context choices between Claude Opus and GPT-4o for k=1 and k=5 documents within the context.

