# ClashEval Supplemental Material

## Dataset Summary

ClashEval is a framework for understanding the tradeoffs that LLMs make when deciding between their prior responses and the contextual information provided.

This Data Card presents information on the ClashEval dataset, which consists of QA pairs accompanied by relevant contextual information. Each question is perturbed along varying degrees. Additionally, the dataset contains questions from six domains:

- Drug dosages
- Olympic records
- Recent news
- Names
- Locations
- Dates

## Dataset Access:
URL to Github: https://github.com/kevinwu23/StanfordClashEval
URL to Huggingface: https://huggingface.co/datasets/kewu93/ClashEval
URL to Croissant metadata record:
https://github.com/kevinwu23/StanfordClashEval/blob/main/data/dataset/croissant.json

## Supported Tasks

question-answering, context-driven generation

## Languages

English

## Data Fields

-`question`: A question that tests knowledge according to one of the six domains provided.

-`context_original`: The original unmodified contextual information that can be used to answer the question.

-`context_mod`: The modified version of the context where the original answer is substituted with the modified answer.

-answer_original: The original unmodified answer to the question.

-answer_mod: The modified answer to the question.

-mod_degree: The degree to which the original answer has been modified. For datasets drugs, news, records, and years, this value is a continuous value corresponding to the numerical change. For names and locations, the values 1, 2, and 3 refer to increasing levels of perturbation according to prompts given in our paper.

-dataset: One of the six domains the question and context are drawn from.

## Licensing Information

CC BY 4.0

## Citation Information

```
@article{wu2024chasheval,
title={ClashEval: Quantifying the tug-of-war between an LLM's internal prior
and external evidence}, author={Wu, Kevin and Wu, Eric and Zou, James},
journal={arXiv preprint arXiv:2404.10198}, year={2024}}
```

## Intended Uses

The data and its analysis are intended for research purposes only.

## Author Statement

We hereby confirm that we bear all responsibility for any violation of rights that may occur in the use or distribution of the data and content presented in this work.

## Hosting and Maintenance Plan

Our dataset is hosted on Huggingface.co and the accompanying code is on Github.