# OpenReview forum: "ClashEval: Quantifying the tug-of-war between an LLM’s internal prior and external evidence"
_NeurIPS.cc/2024/Datasets_and_Benchmarks_Track — NeurIPS 2024 Track Datasets and Benchmarks Poster_

### Official Review · Reviewer_R2pN · 2024-07-17
**ClashEval: Quantifying the tug-of-war between an LLM’s internal prior and external evidence**

**Rating:** 7
**Confidence:** 4
**Correctness:** Yes.
**Clarity:** Yes.

**Review:**

Please see above.

**Strengths:**

1. The dataset is well designed and established. It allows a detailed examination of how LLMs manage conflicting information. As far as I know, such resources are not common.
2. The six LLMs are intensively evaluated. It helps bring deep insight into the strengths and weaknesses of current LLM architectures.
3. This research not only identifies the problem of information conflict in LLMs but also quantitatively measures how deviations in contextual accuracy affect model choices.

**Additional Feedback:**

NA.

**Documentation:**

Yes.

**Ethics:**

No.

**Limitations:**

1. The numerical tests focus on how the six models handle conflicts between their internal prior knowledge and external evidence. It would be better if the author(s) could compare the different performances of these models and discuss the reasons leading to these differences.
2. Why did the author(s) choose the six models for comparison?
3. It would be better if the author(s) could elaborate on how the dataset was created and the experimental setups.
4. The diversity of the dataset's domains and the methods of perturbation seem limited. It may lead the findings of this paper to not generalize well to real-world scenarios.

**Opportunities For Improvement:**

Please see above.

**Relation To Prior Work:**

Yes.

**Summary And Contributions:**

The paper introduces a benchmark dataset called “ClashEval” to assess the ability of LLMs to handle conflicts between their internal prior knowledge and external information retrieved from documents. It explores how LLMs respond to correct or erroneous external content by comparing their internal responses to those influenced by the retrieved context. The study evaluates six leading LLMs. The test results show that they often adopt incorrect information from the context over 60% of the time, particularly when their initial confidence is low.

---

> ### Author Rebuttal · Authors · 2024-08-17
>
> Thank you for your thoughtful feedback! We are pleased to read your positive review and hope to additionally answer some of the comments you have made below:
>
> **Model Choice & Performance**: The models chosen represent top-performing commercial and open-source models according to LMSYS, a popular benchmark, at the time of writing of the paper. Additionally, we include Llama and GPT-3.5 due to their lower costs and the availability of logprobs. To help better understand and compare the model performances, we have also included a per-category model accuracy and (when available) the mean prior probabilities.
>
> **Documentation**: We have included more detailed documentation and example code in our [GitHub repo](https://github.com/kevinwu23/StanfordClashEval):
>
> - `prompts.py`: We have added the prompts we used to generate perturbations on context documents. [Link](https://github.com/kevinwu23/StanfordClashEval/blob/main/data/prompts/prompts.py)
> - `generate_perturbations.ipynb`: We have included a step-by-step recipe for perturbing context documents for each of the dataset types. [Link](https://github.com/kevinwu23/StanfordClashEval/blob/main/generate_perturbations.ipynb)
> - `generate_questions.ipynb`: We also include a step-by-step guide for how we use our question generation prompts to produce QA + context triplets in our dataset. [Link](https://github.com/kevinwu23/StanfordClashEval/blob/main/generate_perturbations.ipynb)
>
> **Data diversity**: We hope to better explain the types of real-world applicability and diversity that exist in our dataset.
> - Domain variety: We cover common knowledge categories that people commonly seek, ranging from sports to news to drug dosages.
> - Information types: We include numerical (drug dosages, sports records), temporal (dates, news events), as well as categorical data (names, locations).
> - Temporal diversity: We also include a mix of historical knowledge (Wikipedia years, names, etc.) to current knowledge (news, sports records). Additionally, we test knowledge on relatively static information (drug dosages, locations).
> - Source diversity: We cover a range of sources, from expert (UpToDate), encyclopedic (Wikipedia), and news (Associated Press).
>
> Additionally, we use a variety of perturbation methods, from numerical (multipliers for dosages, sports records), temporal shifts (dates), and categorical changes (names and locations) which can simulate a variety of misinformation types.

---

### Official Review · Reviewer_NYDi · 2024-07-23

**Rating:** 7
**Confidence:** 5
**Clarity:** Yes.

**Review:**

The paper sheds light on an important property of language models with implications for how often they hallucinate in practical settings. The authors experiment with a fairly wide variety of language models and text domains. The gradated perturbations of incorrect answers is a nice touch. Barring some minor issues with the paper's presentation, I believe it constitutes a solid contribution to the field.

**Strengths:**

- The benchmark is thoughtfully constructed, including data from different text domains and domain-specific answer perturbations
- The authors perform extensive experiments using diverse language models.
- The paper is generally clearly written.
- The benchmark exposes significant deficiencies in state of the art language models.

**Additional Feedback:**

- What is the language model used in Figure 2?
- How are the confidence intervals in Table 2 computed?

**Correctness:**

See above. The benchmark is thoughtfully constructed and the evaluations appropriate (pending some small pieces of missing information).

**Documentation:**

Yes. The collection process is documented in the paper and URLs are available in the supplement.

**Ethics:**

No.

**Limitations:**

The authors adequately address limitations in the conclusion.

**Opportunities For Improvement:**

- First, and most importantly by far, is the fact that per-category accuracy scores/mean prior probabilities are never reported. Given the trends in Figure 4, it seems likely that this information is necessary to interpret the results in Figure 3. The authors claim "Interestingly, these results [in Figure 3] suggest that each model has a different prior distribution over truthfulness across each domain," but, absent this data, it could also just mean that the models have different estimations of their own accuracy in each domain. Is it meaningful, for example, that Claude Opus switches its answer 30% less often than GPT-4o if the latter's original answers are simply less accurate to begin with? In that case, switching answers may be a desirable behavior.
- The authors only run experiments with one document in context at a time. It would be fairly interesting to see how models arbitrate between multiple conflicting sources of information, especially in cases where their prior answers are also present in context.
- The colors in Table 2 are inverted (and are also pretty unnecessary).

**Relation To Prior Work:**

Yes.

**Summary And Contributions:**

The authors propose ClashEval, a new benchmark for measuring how much language models rely on information present in their contexts as opposed to their trained prior. They vary both the domain and the plausibility of in-context hints, and find that state of the art language models are likely to adopt incorrect information presented in-context, especially when the information is outwardly more plausible and/or the language models are unconfident in their prior responses.

---

> ### Author Rebuttal · Authors · 2024-08-17
>
> Thank you for your helpful feedback and positive comments.
>
> We have also added the following as per your review points:
>
> - We have included a per-category accuracy score and mean prior probability. These are also provided in the attached PDF.
> - We have fixed the colors in Table 2.
> - The language model in Figure 2 is GPT-4o. We have updated this in the text.
> - Table 2 confidence intervals are established via bootstrap with N=1000.
>
> **Multi-document context**: We ran additional experiments on GPT-4o and Claude Opus using k=5 documents for each query based on embedding cosine similarity. These results will be available in the appendix. We find that adding more contextual documents lowers overall model accuracy and increases the rate of responses that are neither the prior nor the context. At the same time, due to the lower rate of adherence to context, multi-document RAG also reduces the context bias found in models. These findings are consistent with related works, where models generally perform worse on longer contexts (Levy, et al.) but multiple documents can also protect against hallucination (Xiang, et al.).
>
> Citations:
> Levy, Mosh, Alon Jacoby, and Yoav Goldberg. "Same task, more tokens: the impact of input length on the reasoning performance of large language models." arXiv preprint arXiv:2402.14848 (2024).
>
> Xiang, Chong, et al. "Certifiably Robust RAG against Retrieval Corruption." arXiv preprint arXiv:2405.15556 (2024).

---

> > ### Comment · Reviewer_NYDi · 2024-09-01
> >
> > Thanks for these changes! I strongly support acceptance.

---

### Official Review · Reviewer_ydZz · 2024-07-27
**Evaluating RAG**

**Rating:** 6
**Confidence:** 4
**Clarity:** The paper is easy to follow.

**Review:**

Retrieval-augmented generation (RAG) enhances large language models (LLMs) by providing up-to-date information and reducing hallucinations. However, document retrieval can sometimes introduce incorrect or harmful content. This raises concerns about whether LLMs can recognize and ignore errors or if they repeat them, and whether they can correct their wrong initial responses using retrieved information. To explore this, a dataset of over 1200 questions across six domains (e.g., drug dosages, Olympic records) was created, including content with deliberate errors. Six top-performing LLMs, including GPT-4, were tested. The study found that LLMs adopt incorrect retrieved content over their correct prior knowledge more than 60% of the time. The likelihood of adoption decreases as the retrieved content becomes more unrealistic. Additionally, LLMs are more likely to adopt retrieved information when they are less confident in their initial response.

**Strengths:**

* A benchmark dataset with over 1200 questions across six domains, each accompanied by a relevant contextual document. The answers in these documents are deliberately perturbed to include a range of errors, from subtle to extreme.

* The paper evaluated six top-performing LLMs (GPT-4, GPT-3.5, Llama-3-8b-instruct, Gemini 1.5, Claude Opus, and Claude Sonnet) using this dataset, reporting three key metrics.

* The context preference rates are analyzed across three models, considering both the degrees of perturbation in contextual information and the token probabilities of prior responses.

**Additional Feedback:**

https://github.com/kevinwu23/StanfordClashEval is missing some key artifacts such as code for changing prompts or creating new perturbations. If the authors could point the reviewer to those, the score shall be raised.

**Correctness:**

There is a concern on using logprobs for measuring model's confidence as demonstrated by results showing a lack of calibration in LLMs.

**Documentation:**

The https://github.com/kevinwu23/StanfordClashEval is just a dump of experiments. For example, the raw prompts at https://github.com/kevinwu23/StanfordClashEval/blob/main/data/prompts/prompts.py lack any documentation of how one could regenerate dataset with a different set of prompts. Only a notebook for computing metrics and plotting figures is provided.

**Ethics:**

There are no ethics concerns.

**Limitations:**

One key limitation is that the study is focused on LLMs that are available only via inference, and only one document is being used as context. At the same time, it aims at a systematic exploration of a model’s confidence and the model’s preference for RAG-provided information. The logprobs (or the percentiles used as a form of calibration) provide a very noisy and inaccurate way to measure confidence. So, the current analysis results could be an artifact of logprobs.

**Opportunities For Improvement:**

The generation of dataset needs to be better documented and the released dataset https://github.com/kevinwu23/StanfordClashEval needs to have well-documented means for perturbing prompts or creating new variations.

**Relation To Prior Work:**

This paper builds on the existing exploration of hallucinations, RAG systems, and their evaluation. It also aligns with the study of model confidence and the use of log probabilities to assess this confidence. Unlike prior studies, this work systematically explores the relationship between a model's confidence (via log probabilities) and its preference for RAG-provided information. It also focuses on scenarios where LLMs are available only via inference.

**Summary And Contributions:**

This paper introduces ClashEval, a dataset and benchmark for evaluating LLMs' ability to discern correct from incorrect content/context in the RAG framework. The dataset and evaluations are available at https://github.com/kevinwu23/StanfordClashEval.

---

> ### Author Rebuttal · Authors · 2024-08-17
>
> Thank you for your thorough review of our paper and the provided code. We hope the response below helps answer your concerns.
>
> **Documentation**: As per your feedback, we have included the following additional files in our [GitHub repo](https://github.com/kevinwu23/StanfordClashEval):
>
> - `prompts.py`: We have added the prompts we used to generate perturbations on context documents. [Link](https://github.com/kevinwu23/StanfordClashEval/blob/main/data/prompts/prompts.py)
> - `generate_perturbations.ipynb`: We have included a step-by-step recipe for perturbing context documents for each of the dataset types. [Link](https://github.com/kevinwu23/StanfordClashEval/blob/main/generate_perturbations.ipynb)
> - `generate_questions.ipynb`: We also include a step-by-step guide for how we use our question generation prompts to produce QA + context triplets in our dataset. [Link](https://github.com/kevinwu23/StanfordClashEval/blob/main/generate_perturbations.ipynb)
>
> This additional documentation makes our data more transparent and easier to use.
>
> **Use of Logprobs**: We appreciate the concern about the noisiness of using logprobs. While we acknowledge that token probabilities have limitations as a measure of confidence, we believe several aspects of our results suggest the observed relationships go beyond mere artifacts:
> - We observe a consistent trend between context adherence and token probabilities across different models and question domains, which involve different training procedures and training data.
> - We see practical improvements through the use of token probabilities, which suggest their usefulness as a signal for model confidence.
> - We recognize the potential limitations of raw logprobs and calibration. We attempt to address this by introducing a calibrated version of token probability correction (Section 4.3.1), which also showed improved results.
>
> While these points do not negate the inherent noisiness of logprobs, we believe this baseline method is useful in benchmarking more sophisticated methods.
>
> Thank you again for your time in reviewing our work!

---

> > ### Comment · Reviewer_ydZz · 2024-08-26
> > **Thank you**
> >
> > Thanks for the additional information. I am raising my score.

---

> > > ### Author Response · Authors · 2024-08-30
> > >
> > > Thank you for your time and for your review of our work!

---

### Official Review · Reviewer_SVmZ · 2024-08-06
**A useful, albeit somewhat simplified, dataset for quantifying the clash between the RAG model's internal prior knowledge and the retrieved external content.**

**Rating:** 5
**Confidence:** 3
**Clarity:** I think the paper is clear and well w…

**Review:**

While hallucination of LMs, and using RAG to mitigate this issue, has been investigated extensively in the literature, understanding when an LM adopts erroneous or harmful retrievals appears to have been studied previously less systematically.  This work makes progress in this important direction by studying the relationship between a model's confidence and model's preference for retrieved information.

**Strengths:**

The paper shows that the more erroneous the content is, the less likely the models are to adopt it, and the less confident the model is in its  internal knowledge, the more likely, it is to adopt the external knowledge. The paper also proposes simple calibration techniques based on this insight to improve the accuracy of the models. The paper also provides a dataset that could be used to investigate further questions  around RAG models adopting erroneous or harmful retrievals.

**Additional Feedback:**

N/A

**Correctness:**

Apart from the limitations and suggestions for improvement described elsewhere in this review, the dataset appears to be constructed in a sound way and the evaluation methods and experiment design appears to be appropriate.

**Documentation:**

The dataset is hosted on Hugginface and the code is open sourced on GitHub under permissive licenses. This seems to address the  reproducibility and long-term preservation requirements.

**Limitations:**

Except for the limitations and suggestions for improvement mentioned above, the limitations and potential negative societal impact of this work appears to be adequately addressed.

**Opportunities For Improvement:**

In my view the main limitation is the fact that the ability of LMs to filter errors based on common sense reasoning is not specifically addressed. The errors introduced in the dataset (altered drug dosages, and altered numerical answers to sport records, latest news and Wikipedia years), and many alterations to the Wikipedia names and locations may not necessarily contradict common sense (to the same degree as the generated responses to "eat rocks" or "put glue on the pizza" mentioned in the Introduction would).

Also this work provides only one document for context, which is not generally realistic since typical retrievals involve multiple documents (and possible multiple calls to the external databases).  It would be interesting to understand the issue of erroneous contexts in the end-to-end RAG setting which typically involves multiple retrievals (and possibly discriminating between trustworthy and not trustworthy retrievals).

While the paper suggests calibrating the responses by comparing the token probabilities or percentiles, i.e. choosing the internal response if its probability/percentile is higher that that of the contextualized response, the paper does not address whether it may be possible to fine-tune the models to improve the overall accuracy.

Other limitations (acknowledged in the manuscript) include the fact that the RAG systems can be deployed to domains that are not included in this work, and that the Q&As are fact-based and do not require multi-step logic or other higher level reasoning.

**Relation To Prior Work:**

I believe how this work differs from previous contributions is clearly discussed.  I'm wondering whether the recent TREC RAG dataset (https://trec-rag.github.io) can be leveraged to generate questions and answers in context that require higher-level and/or common sense reasoning.

**Summary And Contributions:**

This work focuses on evaluation of retrieval augmented generation (RAG) when the retrieved context may be erroneous, and more broadly quantifies the tension between the model's internal prior knowledge and the retrieved external content. This work curates a dataset (ClashEval) of 1200+ questions across 6 different domains (drug dosage, news, wikipedia dates, sports records, names, locations) along with content relevant to answering each question. The authors introduce errors to some portion of the answers in the context, and assess 6 leading language models (LMs) on the likelihood of adopting the erroneous content.  (For some questions, the model's internal knowledge is also wrong while the context answers are correct, so the model cannot just ignore the contexts to improve its accuracy.)

---

> ### Author Rebuttal · Authors · 2024-08-17
>
> Thank you for your thoughtful review of our work. We have carefully conducted new experiments based on your suggestions. Please let us know if there are any other questions we can help answer.
>
> 1. **Common sense contradictions**: We appreciate the concern about whether our dataset contains perturbations that contradict common sense reasoning to the degree of examples like "eat rocks" or "put glue on pizza". We strongly believe that the modifications we make (especially extreme perturbations) do produce contradictions along a similar vein and/or magnitude. We will make such examples clear in our paper and we have included a few examples below:
> - Example 1: (Years) For a given question about a film's release date, we modified the date 1994 to 1894, which produces the following GPT-4o response: "The context provided states that the film 'No Escape' was released in the United States in 1894. However, this does not make sense because films as we know them today did not exist in 1894, and 'No Escape' is a modern title that would not fit that time period…"
> - Example 2: (Names) We changed the name of a tennis player "Paul Hanley" to "Waffle McRacketSwing", and GPT-4o produced the following response in GPT-4o: "The name "Waffle McRacketSwing" appears highly unusual and seems unlikely to be the real name of a professional tennis player."
> - Example 3: (Drugs) We changed the maximum daily dosage of ibuprofen from 1,200mg to 12,000mg, which produces the following model response: "The stated maximum daily dosage of 12,000 mg of ibuprofen is extremely high, considering that typical safe maximum daily dosages are around 3,200 mg for prescription-strength ibuprofen."
>
> Our dataset contains many examples of contextual information errors that would be clearly inappropriate for a model to surface to users. At the same time, we want to emphasize that our goal is not just to focus on obvious common sense contradictions but rather to evaluate model behavior across a spectrum of modifications. For example, subtle modifications may pose an even larger risk to users if they are not immediately detectable by common sense. Additionally, in highly expert fields such as medicine or law, the correct answer may actually run contrary to common sense reasoning. We believe a missing puzzle piece is in how models choose between contextual information and parametric knowledge, which is a feature missing from current [common sense reasoning datasets](https://paperswithcode.com/task/common-sense-reasoning/latest).
>
> 2. **Multi-document retrieval**: We agree that it would be interesting to understand how a top-k retrieval setting would influence our results. We ran additional experiments on GPT-4o and Claude Opus using k=5 documents for each query based on embedding cosine similarity. We find that adding more context documents lowers the model's overall accuracy and increases the rate of responses that are neither the prior nor the context. This result is consistent with similar work by Levy, et al. which finds that models perform worse with longer context lengths. At the same time, as a consequence of a lower rate of adherence to context, multi-document RAG also reduces the context bias found in models. This is also consistent with work by Xiang, et al. where they find that using multiple documents can also protect against hallucinations. We have included the results tables for GPT-4o below, and the full tables are in the attached PDF.
>
> **Accuracies: GPT-4o**
> | Dataset    | Acc. Without Context | Acc. With Correct Context (k=1) | Acc. With Correct Context (k=5) |
> |------------|----------------------|---------------------------------|---------------------------------|
> | Drugs      | 0.578                | 0.863                           | 0.819                           |
> | Locations  | 0.575                | 0.925                           | 0.925                           |
> | Names      | 0.445                | 0.990                           | 0.985                           |
> | News       | 0.088                | 0.971                           | 0.924                           |
> | Records    | 0.628                | 0.921                           | 0.911                           |
> | Years      | 0.540                | 0.990                           | 0.990                           |
> | **All**    | **0.467**            | **0.941**                       | **0.922**                       |
>
> **Context and Prior Biases**
> |                      |                    | **Prior Correct**           | **Context Correct**           |
> |----------------------|--------------------|-----------------------------|-------------------------------|
> | **GPT-4o**           | **k=1**            |                             |                               |
> |                      | **Prior Chosen**   | 0.355 (0.321, 0.388)        | 0.041 (0.027, 0.057)          |
> |                      | **Context Chosen** | 0.582 (0.549, 0.617)        | 0.903 (0.881, 0.925)          |
> |                      | **Neither Chosen** | 0.064 (0.048, 0.081)        | 0.056 (0.039, 0.074)          |
> |**GPT-4o**    | **k=5**            |                             |                               |
> |                      | **Prior Chosen**   | 0.535 (0.498, 0.569)        | 0.044 (0.029, 0.060)          |
> |                      | **Context Chosen** | 0.383 (0.349, 0.416)        | 0.868 (0.843, 0.894)          |
> |                      | **Neither Chosen** | 0.082 (0.061, 0.102)        | 0.088 (0.069, 0.111)          |
>
> 3. **Fine-tuning models**: Our main focus is on models in the inference-only API setting, as this is the most common form most commercial LLMs are available today. That being said, we performed a limited set of fine-tuning experiments on GPT-3.5 and found that the model performed around the same or worse. Further fine-tuning experiments on our benchmark is an interesting direction for future work.

---

> > ### Author Rebuttal · Authors · 2024-08-17
> >
> > Thank you for your time in reviewing our work and this response. We hope our follow-up analyses help address the opportunities for improvement.
> >
> > Citations:
> > Levy, Mosh, Alon Jacoby, and Yoav Goldberg. "Same task, more tokens: the impact of input length on the reasoning performance of large language models." arXiv preprint arXiv:2402.14848 (2024).
> >
> > Xiang, Chong, et al. "Certifiably Robust RAG against Retrieval Corruption." arXiv preprint arXiv:2405.15556 (2024).

---

> > > ### Author Response · Authors · 2024-08-30
> > >
> > > Dear Reviewer,
> > >
> > > Thank you again for your valuable feedback on our work. As the discussion phase is nearing its conclusion, we wanted to check if our rebuttal has addressed all your concerns. If there are any remaining questions or if further clarification is needed, please let us know. We are eager to address any points promptly.
> > >
> > > We appreciate you taking your time to review our work.
> > >
> > > Best wishes,
> > > Authors

---

### Decision · Program_Chairs · 2024-09-26

**Decision:**

Accept (Poster)

**Comment:**

The paper proposes a new dataset to examine how external documents can influence LLMs in RAG. It provides interesting observations relating to the confidence of LLMs and the correctness of document contents.
The problem studied is very interesting. The analysis provides new insights on RAG that may inspire follow-up studies.